# Subjective and psychophysiological response to pictures of ancestral and modern threats: Not all evolutionary threats are alike

Iveta Štolhoferová[1☉], Tereza Hladíková[1☉], Markéta Janovcová[1], Šárka Peterková[1], Daniel Frynta[1], Eva Landová[1,2]*

**1** Department of Zoology, Faculty of Science, Charles University, Prague 2, the Czech Republic,
**2** National Institute of Mental Health, Klecany, the Czech Republic

☉ These authors contributed equally to this work.
* evalandova@seznam.cz

**Citation:** Štolhoferová I, Hladíková T, Janovcová M, Peterková Š, Frynta D, Landová E (2026) Subjective and psychophysiological response to pictures of ancestral and modern threats: Not all evolutionary threats are alike. PLoS One 21(3): e0343680. https://doi.org/10.1371/journal.pone.0343680

## Abstract

After encountering a potentially dangerous stimulus, the human body and mind might react with a cascade of physiological, emotional, and cognitive responses to minimize impending harm. However, whether this system can be activated by modern (ontogenetic) threats to the same extent as by ancestral (evolutionary) threats remains uncertain, since the existing results are ambiguous. In this study, we aimed to compare the skin resistance (SR) response to ancestral and modern threats; the focal categories were venomous snakes, heights, airborne diseases, and firearms. We collected recordings of 119 participants, about 30 per threat category, supplemented by participants' rating of the stimuli according to elicited fear. Results showed that participants reacted (SR change) with higher probability to all experimental categories than to control stimuli, with the most frequent reactions to photos simulating the threat of heights, followed by snakes, firearms, and airborne diseases. The largest amplitudes, indicating response intensity, were observed for heights but also for venomous snakes. Further examination showed that higher subjective fear corresponded to an increased probability of SR change. Although the results suggest a slight advantage for ancestral threats, responses to the threat of heights differed in several respects from responses to snakes, demonstrating that ancestrality-based categorisation cannot capture all aspects of the response. Moreover, both ancestral and modern threats can evoke similar electrodermal responses, depending on subjective stimulus salience and/or threat relevance.

## Introduction

Fear is an adaptive emotion that should warn us and prepare us to promptly and adequately respond to threats [1,2]. A perceived danger may stimulate the "fight or flight" response which is regulated by the sympathetic activity of the peripheral

**Data availability statement:** All relevant data are within the paper and its Supporting information files. Moreover, raw data are available via https://doi.org/10.5281/zenodo.17722322.

**Funding:** This project has been supported by the Czech Scientific Foundation (GAČR), project No. 22-13381S, awarded to EL, https://www.gacr.cz/en/. The funder did not play any role in the study design, data collection, analysis, decision to publish, or preparation of the manuscript.

**Competing interests:** The authors have declared that no competing interests exist.

nervous system (PNS) and can be observed, for example, as an increase in heart rate, activation of sweat glands, or pupil dilation. This offers a unique opportunity to measure various physiological responses and investigate the relationship between these objective variables and subjective emotional experience [3,4].

Although fear-evoking stimuli are greatly heterogeneous (e.g., social, environmental/situational, economic, animal, etc.), some stand out due to their long evolutionary significance. According to theories such as biological preparedness [5] and fear module [6], certain stimuli posed a threat to our ancestors, leading to a prepared predisposition for easier acquisition of fear toward those stimuli. In this vein, we may divide various threats based on whether we have shared our evolutionary history with the stimuli or not, categorizing them as ancestral (evolutionarily relevant) or modern (ontogenetic) threats. This distinction is central to the ongoing nature-nurture debate regarding fear processing. The fear module theory proposed that the ancestral threats are processed via a specialized, evolutionarily preserved subcortical pathway capable of triggering autonomic reactions even when the stimuli are presented outside conscious awareness [6]. Strong evidence for this came from studies on subliminal processing, where stimuli are presented outside of conscious awareness (masked). Classic subliminal conditioning studies such as Öhman and Soares [7] demonstrated that fearful individuals exhibit significant skin conductance responses (SCR) to masked images of snakes and spiders, suggesting that ancestral threats can indeed trigger physiological arousal without conscious recognition. Subliminal, unconsciously processed emotions result in an earlier skin conductance response [8]. However, because these emotions are presented very briefly (40 ms, for example) in backward masking paradigm, the physiological response is not fully developed. Conversely, longer-lasting emotional stimuli (of several seconds) that are consciously processed tend to elicit a more intense psychophysiological response (a topic reviewed in [9]). Nevertheless, the manner in which unconscious and conscious threat is processed should differ for ancestral and modern stimuli, resulting in varied psychophysiological responses.

In this study, we aim to compare physiological and emotional responses to ancestral and modern threats represented by four categories of stimuli: (1) venomous snakes, (2) heights, (3) airborne disease, and (4) firearms. Recently, the exclusivity of subcortical pathway for evolutionarily relevant stimuli has been a subject of much debate (e.g., Frumento et al., [10]). In fact, the current state of knowledge suggests that all threatening (fear-evoking) stimuli engage a fundamentally similar brain network that supports their perception and processing and elicits appropriate behavioural responses. This fear network includes the amygdala, thalamus, basal forebrain, insula, anterior and mid-cingulate cortex, prefrontal cortex, and other associated structures [11–13]. Yet despite these shared foundations, different types of threat are processed with substantial variability. This variability arises from the extent to which particular brain regions are activated or deactivated, their functional connectivity and regional homogeneity, and the individuality of the person being examined (e.g., their previous experience; [14,15]).

With regards to the evolutionary versus ontogenetic threats debate, certain trends in fear-network activation can be observed. In an fMRI study, Dhum and colleagues [16] reported stronger activation in several brain regions during presentation of evolutionary versus modern threatening stimuli, including the left inferior frontal gyrus and thalamus, the right middle frontal gyrus, parietal regions, fusiform gyrus, and amygdala. Similarly, Rádlová et al. [17] found heightened activation in the basal ganglia, ventral attention network, and multiple occipital and parietal areas involved in detailed visual perception. In contrast, modern threats elicited greater activation of the posterior cingulate and parahippocampal gyrus [16], as well as the parahippocampal gyrus, angular gyrus, precuneus, and middle temporal gyrus [17]. Surprisingly, these regions are involved in emotional processing, memory encoding and retrieval, and sensory integration. This supports the interpretation of context and the reorientation of attention during a potential threat [18,19]. However, they do not typically initiate the core, immediate 'fight-or-flight' response to threat, which is primarily mediated by the amygdala, hypothalamus and periaqueductal grey, which we would expect precisely in response to ancestral threats. On the other hand, the other two studies focused specifically on the amygdala; Fang et al. [20] supported its stronger activation for evolutionary threats, while Cao et al. [21] showed that presentation of these threats in human versus non-human context may be crucial. While we are unaware of other fMRI studies directly exploring this subject, Wang et al. [22] reported a cognitive processing advantage for ancestral (snakes) compared to modern (guns) stimuli in an electroencephalography study, while Zhang and Guo [23] using event-related potentials, concluded that threat perception is automatic and that phylogenetic threats are more effective than ontogenetic threats in this respect. These studies support the notion that while contextual and personally relevant factors significantly affect neural processing of each threat, there are some systematic differences between how ancestral and modern threat are processed in the brain.

Various other methods have also been used to investigate systematic differences between ancestral and modern threats. Frequently used is visual search task where modern threats often have an advantage. For instance, Blanchette [24] showed that modern threats, such as guns and knives, capture attention in conscious visual search tasks just as effectively as ancestral ones. Studies comparing ancestral and modern threats in masked conditioning paradigms demonstrate that cultural threats can evoke automatic SCR as well, provided the depicted danger is imminent, for example, guns pointed directly at the observer [25]. In these cases, modern threats can survive masking similarly to ancestral threats, suggesting that immediacy and directional relevance may engage rapid defence circuits regardless of evolutionary origin. At the same time, the strength and reliability of these effects remain somewhat greater for evolutionarily ancient stimuli [6], indicating that biological preparedness may still confer a processing bias. Further studies are summarized in a relatively recent review by Shapouri & Martin [26].

However, one otherwise commonly used indicator of sympathetic nervous system activation has scarcely been employed in studies comparing evolutionary and ontogenetic threats. Electrodermal activity (EDA) reflects changes in the electrical properties of the skin due to activity of sweat glands, commonly measured on the palms and soles [27]. Sweating serves not only thermoregulatory and metabolic functions but also occurs during emotional arousal [28], and is under regulation of both the central and autonomic nervous systems. The hypothalamus, the thermoregulation centre, is also sensitive to emotional stimuli. The emotional sweating arises independently of ambient temperature through activation of cholinergic sympathetic fibres [29]. Because the sweat glands respond quickly to emotional or attentional stimuli (1–3 s) [27], EDA provides a sensitive indicator of short-term changes in arousal [4,27,30], allowing researchers to observe event-related physiological reactions. Another advantage of EDA is its relative stability in laboratory conditions, as it is less affected by minor movements or changes in posture or respiration that can interfere with other measures, such as heart rate [31]. In comparison with more technically demanding techniques like facial electromyography (EMG), which requires precise electrode placement on participant's face, EDA is simple to record and non-intrusive for participants. Therefore EDA is particularly suitable for studies investigating immediate, automatic responses to emotionally charged visual stimuli [25,32–34]. A specific parameter of EDA, the skin resistance response (SRR), measures how much the skin impedes electrical current—a value that decreases with increased sweating.

At the break of modern scientific research, authors suggested that in response to a stimulus, it's the physiological changes that come first and this body state is then recognised as an emotion (James Lange Theory; reviewed in [35]). Since then, the view of the connection between subjective experience of fear and the physiological response has shifted: various authors suggested a simultaneous physiological and emotional response (Cannon-Bard Theory; reviewed in [36]), a mediation of emotional response through cognitive appraisal (Schachter-Singer Theory; reviewed in [36]), or even the opposite direction – mediation of physiological response through cognitive appraisal and emotional response (Lazarus's Cognitive Mediational Theory [37,38]). To date, the scientific community has not reached a conclusion (reviewed in [39]). Christopoulos et al. [33] discussed that physiological responses are essential components to emotional experience and might be seen as an unconscious part of the experience that also affect the consciously perceived emotions, hence the connection may lead both ways. When taking a closer look at fear specifically, Taschereau-Dumouchel et al. [40] found that the brain systems responding to emotional evaluation and physiological reactions are rather dissociated, even though generally the subjective fear and the physiological responses were correlated. Thus, it seems that the subjective fear and visceral responses are connected but their relation is more complex than anticipated and that each may affect the other. Moreover, the anxiousness or trait anxiety, i.e., a stable predisposition to perceive a wide array of situations as threatening, might also affect the emotional and physiological response to threats, since people with high trait anxiety are more sensitive to stress and more likely to feel acute anxiety during stressful events [41].

In the current study, we addressed some of these gaps in knowledge and aimed to compare skin resistance responses to ancestral and modern threats in the general (non-phobic) population. We introduced four categories of threats, two representing ancestral perils and two representing modern dangers, to cover a slightly wider portion of the great variability of potential threats (Fig 1). Moreover, we chose not to limit the study to people with extreme emotions toward the selected stimuli but rather to explore the general pattern of physiological reactions in a random population sample. Even though the observed effects might be weaker as a consequence of this sampling method, in our view, this approach fits the evolutionary focus of the study. We further aimed to investigate the relationship between subjective emotional evaluation of the stimuli and objectively measured electrodermal response. Our specific aims were as follows: (1) to compare the skin resistance (SR) change after participants' exposure to images representing various threats; (2) to investigate whether subjectively reported intensity of fear predicts the probability of electrodermal response; or conversely, (3) whether the intensity of physiological response affects subsequently reported intensity of elicited fear; (4) to test whether participants' general anxiousness affects the probability of electrodermal response to any type of stimulus and (5) whether participants' sensitivity to a particular type of threat affects the probability of electrodermal response to that threat.

## Methods

### Selection and preparation of the stimuli

To compile the experimental stimuli sequences, we started with a selection of four sets of 28 threat images and a set of 30 control images. The photos were sourced from image databases, including OASIS [42], IAPS [43,44], SMID [45], EmoMadrid [46], Pixabay, Wikimedia Commons, and Flickr, or were taken by the study authors (see S1 Table for details).

All threat images represented one of four distinct types of threats: (1) venomous snakes, (2) heights, (3) airborne diseases, or (4) firearms. Theoretical background for stimuli categories selection is provided in S1 File. For venomous snakes, we chose 28 species of snakes from the family Viperidae (i.e., vipers, adders, and rattlesnakes) since the viperid morphotype was consistently shown to elicit the greatest fear [47–49]. The photos depicted snakes in their natural habitats and non-attack body postures, the whole head and body were visible. The snake's head was oriented toward the camera lens or to the side.

For heights, we focused on photos with a downward perspective to emphasize the depth and to better evoke the feeling of danger. We favoured photos of natural scenes (e.g., cliffs, crags) but did not avoid man-built structures (e.g., buildings, bridges) either. Several photos also depicted people (small figures in the distance or shoe tips at the bottom of the frame) to enhance the depth perception and to figuratively place the participant in the situation.

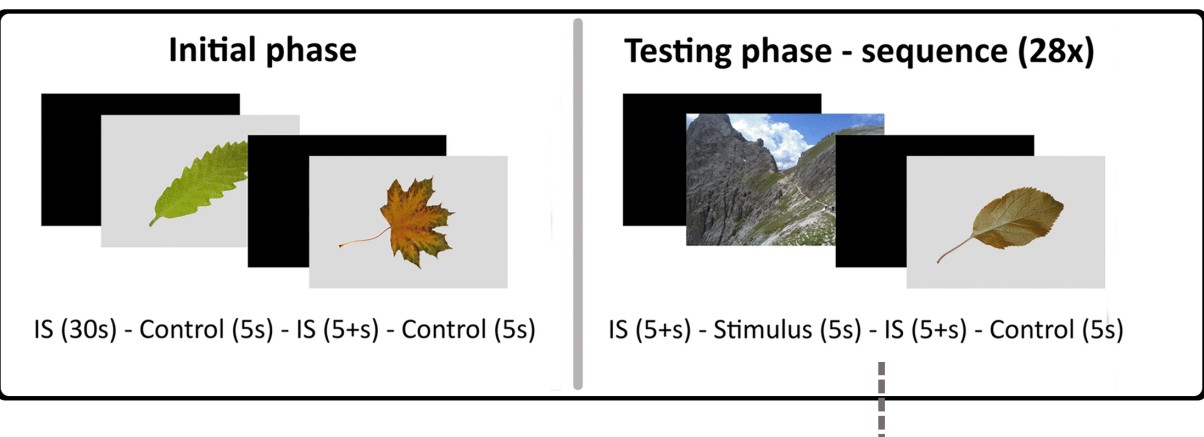

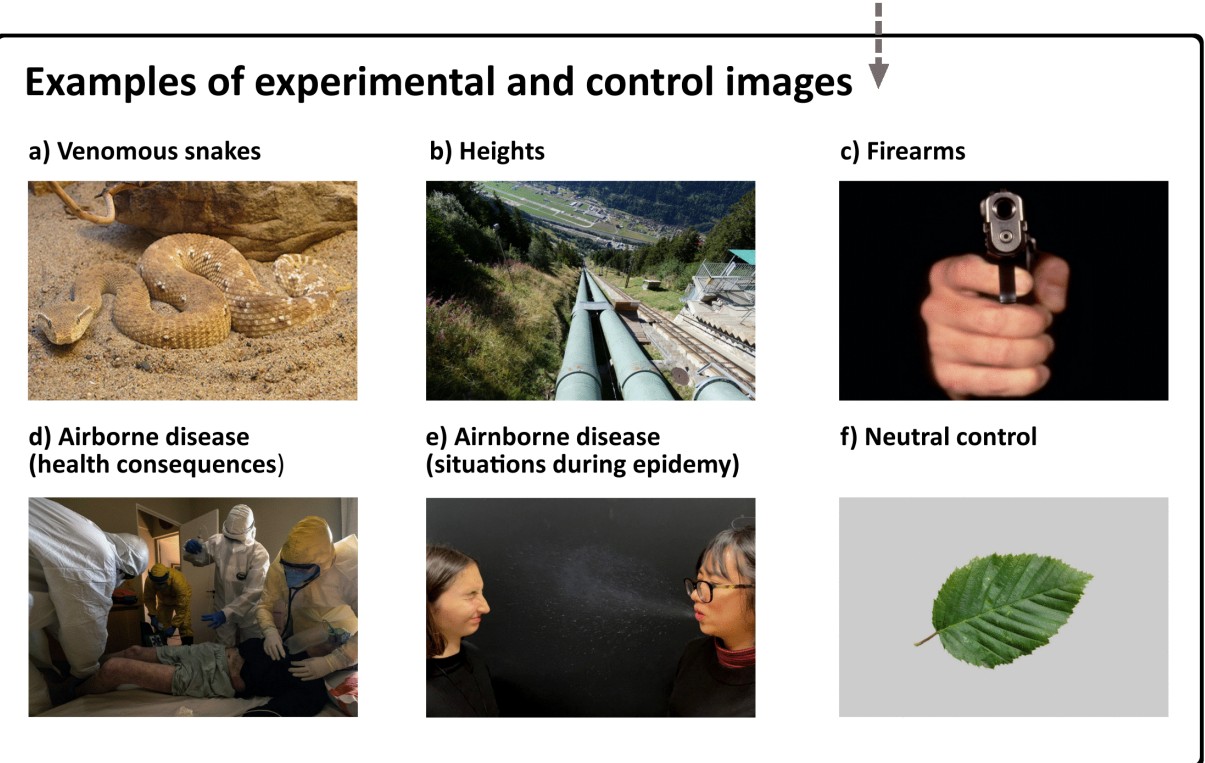

**Fig 1. Design of the experiment and examples of experimental and control stimuli. IS stands for interstimulus (black screen).**

For airborne diseases, half of the photos depicted everyday-life situations during the epidemy of an airborne disease (people in public spaces wearing masks properly or under their nose, social distancing or crowding, sneezing, coughing) evoking the fear of contamination. The other half aimed to evoke fear of the serious health consequences of the disease through photos of medical staff attending patients in IC (intensive care) units while wearing masks, suits, and other protective equipment.

For firearms, we chose a variety of photos depicting a person (man, woman, or child; civilian, or soldier) holding a gun. The person was either identifiable (face and body were shown) or not (the photo depicted only hands). The guns were pointed toward the camera lens, toward another depicted person, or an unclear (not shown) target away from the camera lens. Guns were either short (e.g., revolvers) or long (assault rifles). Five photos depicted guns outside of human context, e.g., lying on a table.

If needed, the experimental images were cropped to the required size of 1772 x 1181 pixels; otherwise, the photos were not modified. For control stimuli, we chose photos of various leaves based on previous research showing that leaves are not fear-eliciting but rather emotionally neutral stimuli [48]. The original photo background was cut off and a single leaf was placed in the centre of a uniform 20% grey background [48].

## Testing procedure

The participant was invited to the testing room, briefed about the experimental procedure, and signed a written informed consent. Next, the participant was seated in front of the testing screen (approx. 60 cm apart) and the experimenter attached the sensors for measuring electrodermal response. We used Multifunction Biotelemetry Support System for Psychophysiology Monitoring VLV3 [50], which enables measuring the response in real time during the stimuli presentation [48]. Skin resistance (SR) was measured using dry sensors attached to the distal phalanx of the index and middle fingers of the non-dominant hand. The participant was asked to place their hands on the table and watch the whole stimuli sequence avoiding unnecessary movements to minimise measurement artifacts.

The stimuli were presented on 26-inch, Quad HD (2560 x 1440 pixels) screen. Each experimental sequence consisted of threat stimuli of only one type while control stimuli were the same in all experimental sequences. The threat and control stimuli alternated in a semi-randomized order, ensuring that each threat stimulus was followed by a control stimulus and vice versa. Each stimulus was displayed for exactly 5 seconds followed by black screen (interstimulus) presented for at least 5 seconds until the participant's SR signal had flattened and approximately returned to baseline (Fig 1). The experimenter continuously monitored the SR signal in real time to ensure this criterion was met before the next stimulus appeared. This procedure minimized the possibility of anticipatory responses. To further reduce predictive processing effects, responses that peaked within the first second after stimulus onset were excluded from analysis, as SR changes typically occur only after approximately 1000 ms; earlier responses likely reflect anticipation rather than stimulus-evoked activity [27]. The very first two stimuli were controls to familiarize the participant with the procedure; reactions to these stimuli were dropped from further analyses. Each sequence was presented in one of three fixed orders; the procedure lasted for approx. 15 minutes. The experiment followed a between-subjects design. In total, 30 participants were presented with a venomous snakes experimental sequence, 31 saw a heights experimental sequence, 30 watched a firearms experimental sequence, and finally 30 viewed an airborne diseases experimental sequence. However, results of two participants (one airborne disease and one firearms session) were excluded due to technical difficulties during data extraction leading to a total of 119 analysed experimental sequences. The experimental sequence and its variant were chosen at random for each participant without experimenter's *a priori* knowledge of participant's attitude toward the tested stimuli category.

After the physiological measurement, the participant was asked to rate the just-presented threat stimuli on 7-point Likert scale [51] according to the intensity of elicited fear. Rate 1 stood for no fear, while rate 7 represented extremely high fear. Lastly, he or she completed several psychometric questionnaires (see below for details).

## Participants

All participants, 119 in total, were adult Czech, Slovak, or other Central Europeans, aged 18–74 (median = 23, mean = 29.78, SD = 14.56), although 8 participants chose not to report their age. Since women are more sensitive to some negative emotions including fear (McLean & Anderson, 2009), our study included more women than men: 23 (19.3%) participants were men and 96 (80.7%) were women. Participants were actively recruited via poster fliers, promoting the research on Facebook sites of involved institutions, and by advertising to former participants of our previous projects.

To further characterize the participants, they were asked to fill in the following questionnaires: (1) State-Trait Anxiety Inventory-X2 (STAI-X2) [52,53], (2) The Three Domains of Disgust Scale (TDDS) [54,55]; we modified the questionnaire

by excluding the items of sexual domain, (3) Coronavirus Safety Behaviours Scale (CSBS) [56]; originally developed in the context of Ebola epidemy [57], (4) The Fear of COVID-19 Scale (FCV-19S) [58,59], (5) The COVID Stress Scales (C-19SS) [60]. Moreover, participants filled in one additional questionnaire concerning the fear of stimuli they were shown: (1) Twelve-Item Snake Questionnaire (SNAQ-12) [61–63], (2) Acrophobia Questionnaire (AQ) [64], or (3) Attitudes Toward Guns and Violence Questionnaire (ATGV) [65,66]; we used modified version more suitable for Czech environment, see S2 File). All questionnaires were in Czech; if published translation was not available, we translated the questionnaire ourselves. Some of the completed questionnaires also supplemented dataset of another unrelated study [Polák et al., under review]. Unfortunately, several participants chose not to fill in/complete some questionnaires which slightly affected the number of available datapoints for Spearman correlations. Since the N was different for each questionnaire, it is reported in the Results section together with the correlation coefficients.

## Data extraction and curation

We analysed 119 sequences with 56 (28 experimental + 28 control) stimuli per sequence leading to a total of 6,664 measurements of physiological reactions. Nonetheless, 167 measurements (2.51%) were excluded due to moving participants, making it ambiguous whether the observed reaction was an emotional reaction to the stimulus or a direct consequence of the motion.

The physiological reaction was further analysed through two noted variables (Fig 2): presence of an electrodermal reaction (i.e., a SR change curve; 0–1 binary variable), and amplitude of the SR change curve (in kiloohms, kΩ). We further measured the latency of the reaction (from the stimulus exposure to the beginning of the SR change curve; in seconds) which was not directly analysed but was used for data curation: Only SR change curves with latency between 1–5 seconds were considered as emotional reactions to the presented stimulus. The lower threshold reflects the implicit temporal delays such as time needed for processing of the stimulus, autonomic nervous system nerve conduction to the sweat glands, or penetration of sweat to the epidermis. Conversely, the upper threshold was set for optimal detection of an immediate emotional response rather than a response to complex cognitive processing of the stimulus (see [27] for detailed discussion).

## Statistical analyses

Full datasets associated with this study are available as S2 and S3 Tables. To determine whether stimuli categories differ in the probability and amplitude of skin resistance (SR) change, we implemented generalized mixed-effect models (GLMM) as built in RStudio 4.3.1 (R Core Team, 2023), package lme4 [67]. For the probability of SR change, we built GLMM model of binomial family (logit link) with the presence of an SR change as response, stimuli category as fixed factor, and participants' ID as random factor. For detailed analysis of observed physiological reactions, we dropped null reactions and analysed only cases where SR change was observed. We built a separate GLMM model of Gamma family (natural logarithm link) with the amplitude of the SR change as response and LMM model with latency of SR change onset as response. Both models used stimulus category as fixed factor, and participants' ID as random factor. Moreover, we used GLMM model to investigate whether the probability of SR change is affected by subjectively reported fear of the stimulus, employing Likert scale fear rating, stimulus category, and their interaction as fixed factors. The models were tested against null models (i.e., models with no fixed factors) and the best models were chosen based on the Akaike information criterion (AIC). Effect sizes were estimated with package performance [68] with marginal $R^2$ reflecting variance explained by fixed effect and conditional $R^2$ reflecting variance explained by both fixed and random effect. Following Lorah's [69] interpretation of marginal $R^2$, we considered effect size of $R^2 = 0.02$ to be small, $R^2 = 0.13$ medium, and $R^2 = 0.26$ to be large.

We also employed multivariate methods, k-clustering specifically, to investigate whether individual stimuli would cluster in their pre-defined categories based on assessed parameters of skin resistance change alone. This approach did

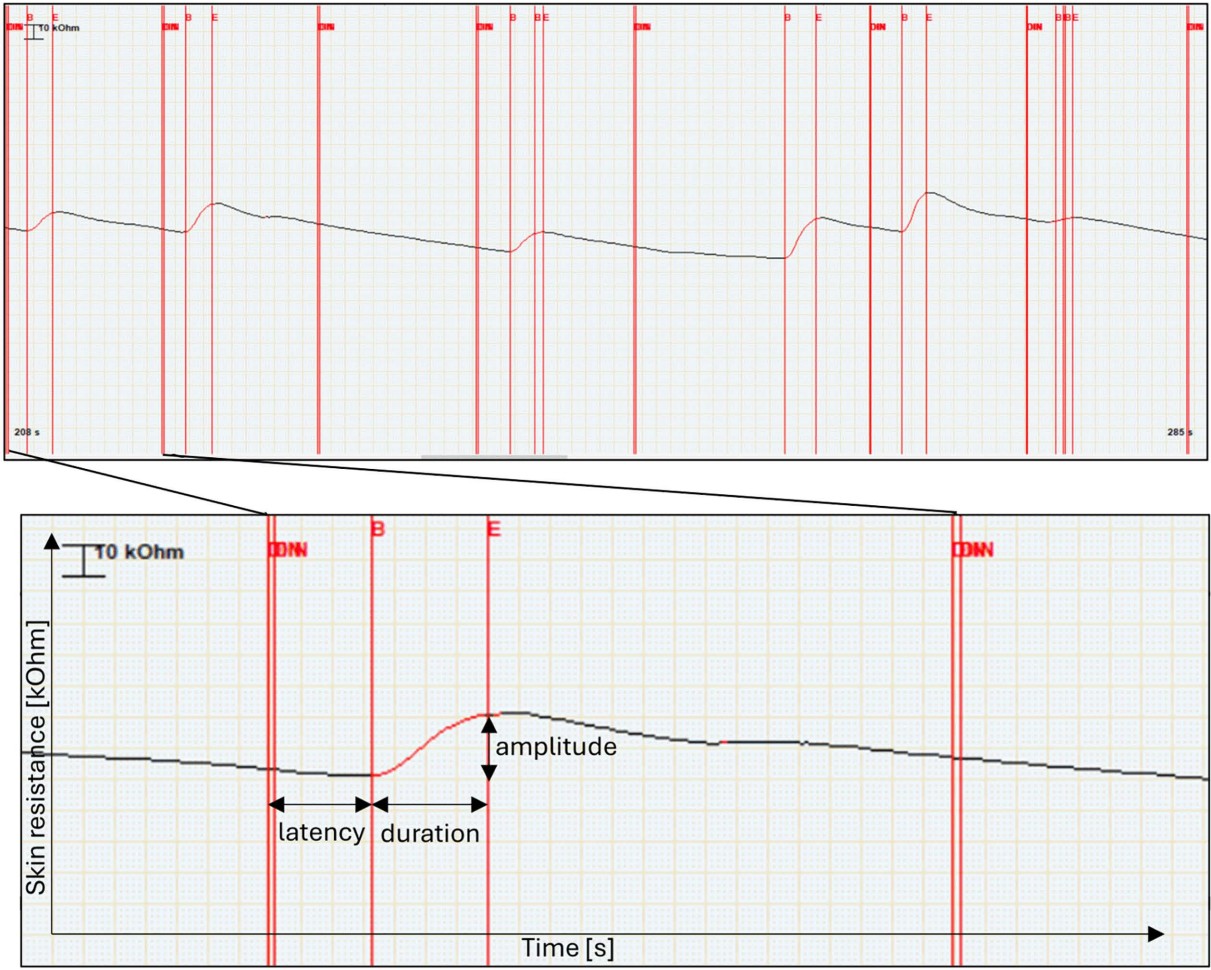

**Fig 2. Physiological recording of skin resistance – an example from data.** The upper panel shows a continuous segment of skin resistance recording measured in kΩ. Vertical double red lines (DIN) indicate the start of stimuli, dividing the segment into seven epochs. Vertical single red lines indicate the onset (B) and maximum (E) of the skin resistance response. The lower panel provides a zoomed-in view of a single response, illustrating the key parameters measured: latency (time from stimulus onset to the onset of the response), duration (time from the response onset to the response maximum), and amplitude (difference between maximum and minimum within the peak).

not prove very powerful perhaps simply because too few parameters of SR change could inform the analysis. Detailed description of this method as well as its results can be found in S3 File.

To assess the effect of experienced electrodermal response (observed SR change) on subsequent rating of the stimuli, we employed cumulative link mixed models (CLMM) as implemented in the ordinal package [70], which are specifically designed for ordinal dependent variables such as Likert-scale ratings. The model converts raw ratings into probability of each particular rating (1–7) using a logit function. The output includes threshold coefficients that indicate the cumulative probabilities for each rating. For instance, if the threshold coefficient for 3|4 is 0.65, the probability of receiving a rating of 3 or lower is 65%. The models can also evaluate the effects of fixed factors on the threshold coefficients. Here, we used Likert-scale ratings as response, presence/absence of SR change, stimulus category, and their interaction as fixed factors, and respondents' ID as a random factor.

Finally, to assess the relationship between general anxiousness, sensitivity to specific threats, and the degree of electrodermal response, we computed Spearman rank corelations. General anxiousness was represented by the score of STAI-X2 questionnaire, sensitivities to investigated types of threats were represented by scores of other questionnaires listed above. The degree of electrodermal response was computed as mean amplitude of SR change across all stimuli per participant and across experimental stimuli only (control stimuli excluded). We utilized Spearman rank corelation to adjust for non-normal distributions of the questionnaire scores.

### Ethical note

All procedures performed in this study were carried out in accordance with the ethical standards of the appropriate institutional research committee, the Ethical Committees of Charles University, Faculty of Science (approval no. 2021/02, granted on 14 April 2021) and National Institute of Mental Health (no. 91/21, granted 31 March 2021), and with the 1964 Helsinki declaration and its later amendments or comparable ethical standards. Written informed consent was obtained from all participants included in the study.

## Results

### Physiological reactions toward stimuli of different categories

The alternative model for probability of physiological reaction with stimuli category as fixed factor was significantly better than null model (i.e., model with no fixed factors): $AIC_0 = 6365$, $AIC_A = 6138$, $\chi^2 = 234.62$, $p < 0.001$, marginal $R^2 = 0.059$, conditional $R^2 = 0.334$. Detailed examination of the model showed that participants (within-subject comparison) reacted with greater probability to all experimental categories than to control stimuli (leaves): heights – prob. = 0.420, $z = 13.43$, $p < 0.001$; venomous snakes – prob. = 0.203, $z = 5.10$, $p < 0.001$; firearms – prob. = 0.188, $z = 3.94$, $p < 0.001$; airborne diseases – prob. = 0.156, $z = 2.14$, $p = 0.032$; control – prob. = 0.122 (Fig 3a). Between-subject comparison of different types of threats showed that the probability of reaction to heights was significantly higher than to all other types of threats ($p < 0.001$). Otherwise, there was no significant difference between any other threat categories ($p > 0.05$; detailed results in S4 Table).

Likewise, the alternative model for amplitude of the SR change curve with stimuli category as fixed factor was significantly better than the corresponding null model: $AIC_0 = 10801$, $AIC_A = 10735$, $\chi^2 = 73.53$, $p < 0.001$, marginal $R^2 = 0.091$, conditional $R^2 = 0.360$. However, compared to control stimuli (within-subject comparison), the amplitude of the physiological reaction differed only for heights and venomous snakes but not for firearms or airborne diseases: heights – ampl. = 20.74 kΩ, $t = 8.72$, $p < 0.001$; venomous snakes – ampl. = 12.24 kΩ, $t = 2.59$, $p = 0.010$; firearms – ampl. = 11.23 kΩ, $t = 1.35$, $p = 0.178$; airborne diseases – ampl. = 9.80 kΩ, $t = 0.01$, $p = 0.989$; control – ampl. = 9.79 kΩ (Fig 3b). Similarly to probability of physiological reaction, between subject comparison showed that the amplitude of the SR change curve was significantly the highest for heights stimuli ($p < 0.001$) but otherwise, the threat stimuli categories did not differ ($p > 0.05$; detailed results in S4 Table).

### Temporal course of SR change

Similarly to response probability and amplitude, we also examined latency of SR change onset relative to the onset of stimulus. Although the category effect was statistically significant, its size was very small: $AIC_0 = 814.2$, $AIC_A = 810.4$, $\chi^2 = 11.78$, $p = 0.019$, marginal $R^2 = 0.009$, conditional $R^2 = 0.096$. Moreover, the only category that differed from the control was airborne disease with significantly shorter latencies of SR onset change than control: airborne diseases – lat. = 2.36 s, $t = -3.07$, $p = 0.002$; firearms – lat. = 2.53 s, $t = -1.41$, $p = 0.159$; heights – lat. = 2.54 s, $t = -1.63$, $p = 0.103$; venomous snakes – lat. = 2.57 s, $t = -1.19$, $p = 0.233$; control – lat. = 2.68 s. Fig 4 shows frequency of observed SR changes onsets per stimulus category during eight 500ms intervals of the analysed period.

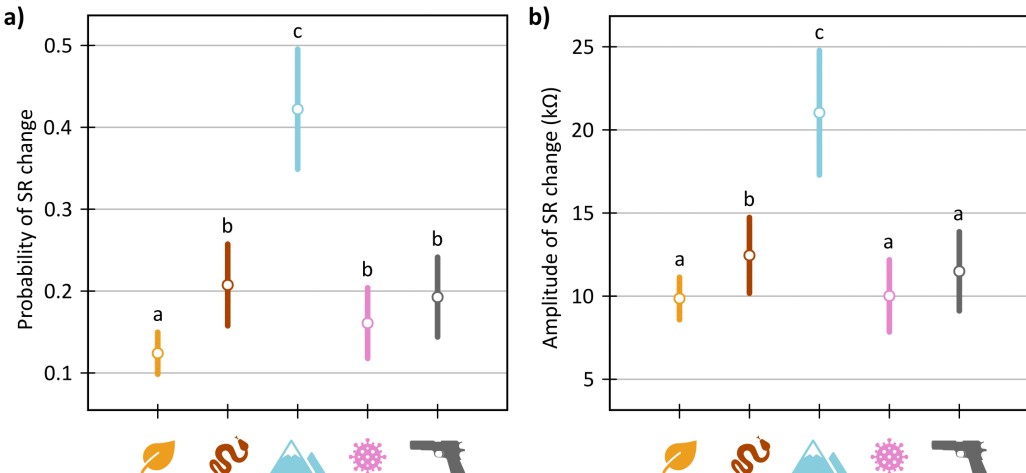

**Fig 3. Mean and 95% confidence interval of probability of SR change (a) and amplitude of SR change (b) per each category of stimuli.** While the probability of electrodermal response convey that changes of skin resistance was larger for all experimental categories than for control stimuli (leaves), the amplitude of SR change suggest superiority of evolutionary threats. Nonetheless, both parameters clearly indicate that the strongest responses were measured for heights stimuli, even in comparison with venomous snakes stimuli. Same letter indexes indicate unsignificant difference between the categories. The pictograms correspond 5 investigated stimuli categories: leaves, venomous snakes, heights, air-borne diseases, and firearms.

## Effect of subjective fear on physiological response

Out of the four threat categories, venomous snake stimuli tended to be subjectively rated as eliciting the highest fear, while air-borne diseases as eliciting the weakest fear: venomous snakes – mean=3.41, median=3, SD=1.95; heights – mean=2.79, median=2, SD=1.84; air-borne diseases – mean=2.57, median=2, SD=1.96; firearms – mean=3.16, median=3, SD=1.96. For histograms of subjective rating of stimuli per category, see S1 Fig. To investigate whether the probability of physiological reaction is affected by subjectively reported fear of the stimulus, we built GLMM model with Likert scale fear rating, stimulus category, and their interaction as fixed factors. This model proved the best (AIC=3357.1) compared to simpler model without the interaction (AIC=3360.3, p=0.027), or the null model (AIC=3401.1, p<0.001). The model also explained relatively substantial amount of variability: marginal $R^2$=0.071, conditional $R^2$=0.368. Detailed examination of the model further revealed that higher subjective fear increased probability of physiological reaction in heights (z=5.53, p<0.001), firearms (z=2.14, p=0.033) and airborne diseases (z=2.12, p=0.034) but not in venomous snakes (z=0.91, p=0.363). Detailed results are shown in Fig 5 and S5 Table.

## Effect of physiological response on subjective fear

To investigate whether the presence/absence of physiological reaction affects subjectively reported fear of the stimulus, we utilized cumulative link mixed models (CLMM) with Likert scale fear ratings as response, presence/absence of SR change, stimulus category, and their interaction as fixed factors, and respondents' ID as random factor. This model proved the best (AIC=10,428) compared to simpler model without the interaction (AIC=10,439; p<0.001), model with presence/absence of SR change as the only fixed factor (AIC=10,441; p<0.001), model with stimulus category as the only fixed factor (AIC=10,473; p<0.001), and finally the null model (AIC=10,474; p<0.001). Detailed examination of the model showed there was a significant difference in elicited fear rating between the trials when the SR change was measured and trials when it was not for heights stimuli (z=6.36; p<0.001), firearms stimuli (z=2.60; p=0.009), and air-borne diseases stimuli (z=2.17; p=0.030) but not in venomous snakes stimuli (z=0.73; p=0.466; Fig 6). Detailed results are shown in S6 Table.

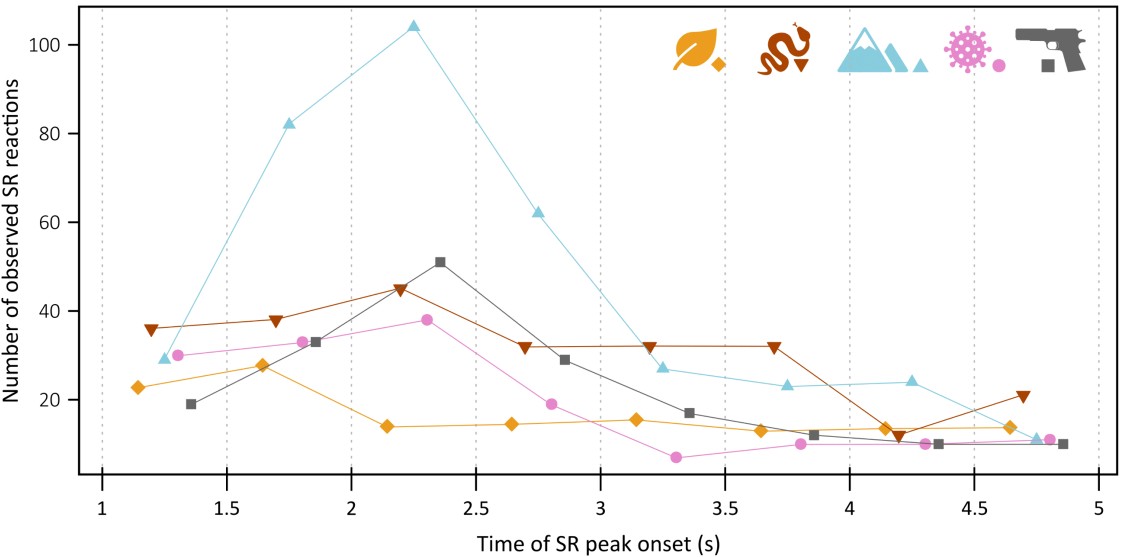

**Fig 4. Frequency of observed onset latencies of skin resistance changes for each stimulus category.** The analysed window from 1 to 5 seconds after stimulus onset was divided into eight 500 ms intervals. Raw frequencies are shown, except for the control category, which was divided by four because control stimuli were presented to four times as many participants as any threat category; this adjustment facilitates comparison across categories. The pictograms correspond 5 investigated stimuli categories: leaves, venomous snakes, heights, air-borne diseases, and firearms.

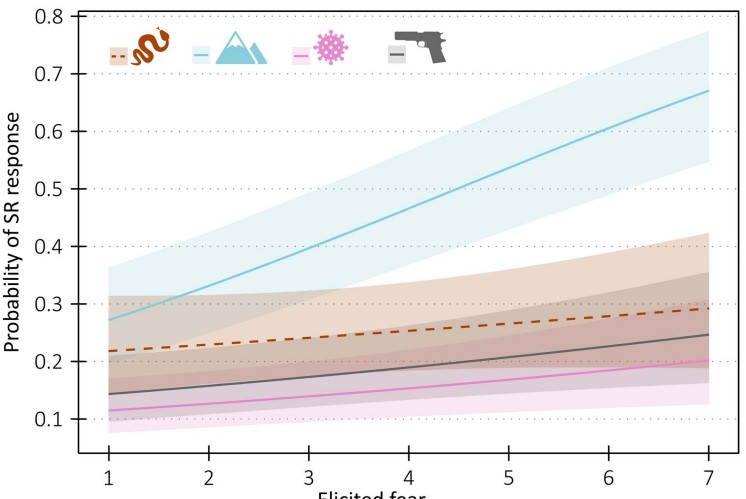

**Fig 5. The effect of subjectively reported elicited fear on probability of SR change for each category of experimental stimuli, ribbons around the lines represent 90% confidence intervals.** Note that only solid lines significantly differ from zero, i.e., fear elicited by the stimuli affected the probability of electrodermal response only in heights, firearms, and airborne disease stimuli but not in snakes (dashed line). The pictograms correspond the experimental stimuli categories: venomous snakes, heights, air-borne diseases, and firearms.

## Relationship between general anxiousness and physiological response

Next, we tested whether participants' general anxiousness (measured by STAI-X2 questionnaire) affects the amplitude of SR change. To this end, we computed average amplitude of the SR change curve, including null values, for each participant across all stimuli and across threat stimuli only (control stimuli excluded). However, neither variable was significantly

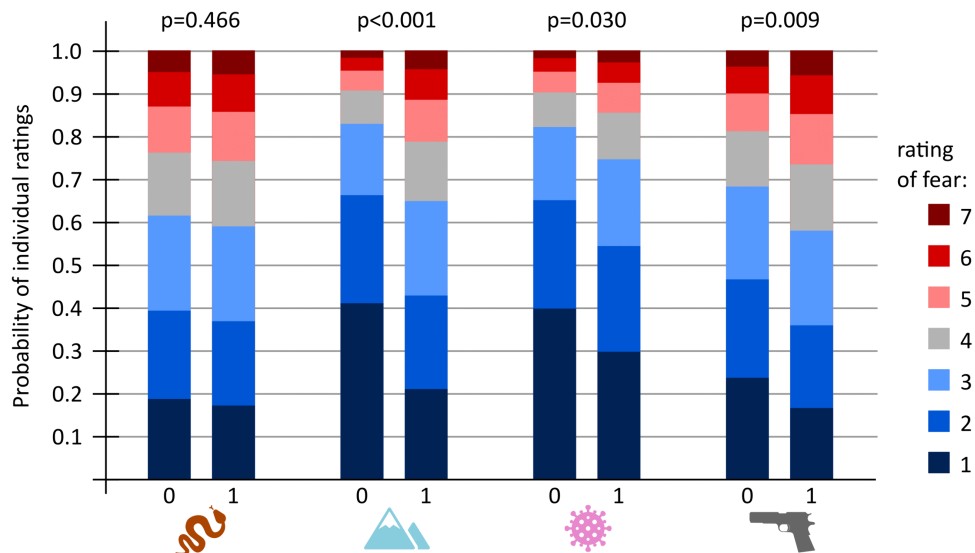

**Fig 6. The effect of observed SR change on the probability distribution of fear ratings for each experimental stimulus category.** Each colour corresponds to a subjective rating on the 7-point Likert scale, from dark blue representing no elicited fear (rating = 1) to dark red representing very high elicited fear (rating = 7). The height of each coloured segment indicates the probability that a given rating was reported. Columns marked "0" represent trials in which no SR change was observed, whereas columns marked "1" represent trials in which an SR change was measured. Above each pair of columns are p-values for the null hypothesis that there is no difference in fear ratings between trials with and without an SR change. For stimulus categories with significant effects (heights, air-borne diseases, firearms), SR change is associated mainly with a reduction of stimuli rated as 1 ("no fear") and an increase in the number of highest ratings (5–7). Intermediate ratings (2–4) represent approximately the same portion of cases in both types of trials (with and without observed SR change). The pictograms correspond the experimental stimuli categories: venomous snakes, heights, air-borne diseases, and firearms.

correlated with STAI-X2 score: correlation with all stimuli – N = 106, $R_{Spearman}$ = −0.161, p = 0.100; correlation with threat stimuli only – N = 106, $R_{Spearman}$ = −0.142, p = 0.146.

### Relationship between sensitivity to specific threat and physiological response

Finally, we examined whether participants' sensitivity to particular type of threat affects the amplitude of SR change. Similarly to above, we computed average amplitudes of the SR change curve, including null values, for each participant and performed Spearman correlation with questionnaire scores (or sub-scores) relevant to the stimuli they were presented with. Since the vast majority of correlations did not significantly differ from zero, below, we report only the best correlated questionnaire/questionnaire subscale per category. Full results are available in S7 Table.

The only significant correlation was found for venomous snake stimuli. Although the SNAQ-12 score was not significantly correlated with the average amplitude of SR change for all stimuli (N = 27, $R_{Spearman}$ = 0.335, p = 0.090), it was significantly and moderately correlated with the average amplitude of SR change for threat stimuli only: N = 27, $R_{Spearman}$ = 0.424, p = 0.027.

Regarding the height stimuli, we found no significant correlation between avoidance sub-scale score of AQ with either the average amplitude of SR change for all stimuli (N = 30, $R_{Spearman}$ = −0.246, p = 0.190) or with threat stimuli only (N = 30, $R_{Spearman}$ = −0.204, p = 0.279).

Concerning the air-borne diseases stimuli, there was no significant correlation between CSBS score and the average amplitude of SR change for all stimuli (N = 29, $R_{Spearman}$ = 0.335, p = 0.090) or for threat stimuli only (N = 29, $R_{Spearman}$ = −0.142, p = 0.146).

Finally, as for the firearms stimuli, the score of perceived danger subscale of the modified ATGV questionnaire was not significantly correlated with the average amplitude of SR change for all stimuli (N = 22, $R_{Spearman}$ = 0.011, p = 0.962) or for threat stimuli only (N = 22, $R_{Spearman}$ = 0.239, p = 0.284).

## Discussion

In this study, we aimed to compare psychophysiological reactions to different types of threats with relation to their relevance for human evolutionary history. To this end, we compared skin resistance (SR) responses to four types of threats: venomous snakes, heights, air-borne diseases, and firearms. Even though we cannot pinpoint the exact period the investigated types of threats have represented a significant selection pressure to humans or their ancestors, the inconsistent, often uniform pattern of SR response to all of them suggests against the simple distinction between ancestral and modern threats. To name a particular example, SR response to venomous snakes and firearms did not significantly differ in the probability of SR change despite the evolutionary relevance of these two types of threats certainly being very different (Fig 3a). This clearly indicates that the threat's "co-evolution" period is not the sole factor we should consider.

These results are not completely new since a quick review of the relevant existing literature suggest that the effect of "co-evolution" period is ambiguous at best. Several studies have already explored the effect of snakes and guns on fear conditioning [25,71–73] (see also review [74,75]), attention (e.g., [76–79]; reviewed in [26]), or by employing neuroimaging [16,20,21,23,80]. As the recent review of Shapouri and Martin [26] concludes, studies comparing ancestral and modern threats (mostly represented by snakes and guns) have not brought conclusive evidence of superiority of either of these types of stimuli which in itself warrants against superiority of ancestral threats as predicted by the fear module theory [6]. The review's authors (Shapouri and Martin [26]) suggest to approach the topic from the view of relevance detection hypothesis, a wider concept originally formulated by Sander and his colleagues [81]. Sander et al. [81] proposed that the extensive involvement of amygdala is not limited to processing of fearful stimuli but rather to a wider range of biologically and socially relevant stimuli, including the pleasant ones. Although generally relevant suggestion, here, we will further limit ourselves to discussion of the role of fear. This is not due to us favouring the fear module theory but rather due to the highly significant effect of subjectively reported fear on SR change responses.

Even today, there is not one agreed-upon viewpoint of what "fear" is and, after us encountering a dangerous stimulus, where does this emotion we label as fear come from [82]. Although we did not address this issue specifically during the experiment, we analysed the same data twice – firstly, we assessed the effect of subjectively reported fear on SR change, secondly, we assessed the effect of SR change on subjectively reported fear. Both methods gave similar results: the relationship between the SR change and subjective fear proved significant for heights, firearms, and air-borne diseases, and null for venomous snakes. This was surprising to us since the link between subjectively perceived fear of snakes and objectively measured skin resistance change was anticipated based on repeatedly reported larger electrodermal response of fearful (ophiophobic) than of low-fear individuals [7,48,83,84]. In our previous study, for example, the average amplitude of SR change and average fear rating of the snake stimulus were highly correlated with $R_{Spearman}$ = 0.773 [48]. Admittedly though, other studies did not find any difference between these groups of participants [85,86]. Moreover, the diversity of snake stimuli in the previous study [48] was much greater; the study included also non-dangerous and rather disgust-eliciting snakes [87] which affected this particular result. As we showed in several of our previous studies, people distinguish between several snake morphotypes, some of which, namely viperids, consistently and even cross-culturally evoke greater fear then others [47,49,88,89]. We hypothesize that the unconscious processing of the viperid stimuli affected the physiological response to a greater degree than conscious processing. This idea comes from the results of a thorough electrodermal study by Tan and colleagues [90] where animate negative stimuli consistently elicited stronger skin conductance response then non-animate negative objects when presented in unconscious conditions (masked, for 40ms). Conscious processing is, on the other hand, reflected more directly in the Likert scale ratings which caused the discrepancy between the two responses. As Manassero and colleagues [91] showed, implicit and explicit processing

may be dissociated in threat evaluation. In their conditioning experiment, implicit responses were rooted in detection of a specific cue while explicit responses were based on wider generalization patterns. This dissociation mirrors the "nature vs. nurture" debate regarding subliminal processing. As demonstrated by Öhman and Soares [7], ancestral threats like snakes can trigger physiological arousal via subcortical pathway, independent of conscious identification. In our study, this automatic response (SR) may have occurred distinct from the conscious or cognitive appraisal (subjective rating), leading to the observed lack of correlation.

Nonetheless, specific sensitivity to fear of snakes assessed through the SNAQ-12 score can be used as a good predictor for the degree (average amplitude) of SR change after exposure to snake stimuli. This result is in line with the previously cited studies [7,48,83,84], here further supported on a wider sample of general population, not two contrasting groups of low-fear and high-fear individuals. Finally, we demonstrated a complete lack of correlation between the general anxiousness and the SR change response to any type of experimental or control stimuli. This is an important finding illustrating that the results of SR change measurement are not biased towards more generally anxious subjects which supports it as a suitable method for assessment of psychophysiological reaction in a wide variety of respondents.

In contrast to venomous snakes, we found a strong association between SR response and conscious fear rating in stimuli simulating the threat of heights (e.g., Fig 5). This is in accordance with most relevant studies that generally show increased skin conductance and heart rate after exposure to a challenging height-related scenario in virtual reality [92–97], although several studies did not find any electrodermal response [98–100]. Moreover, Bzdúšková et al. [92] found significantly stronger electrodermal responses to height scenario in high-fear compared low-fear participants, although Diemer et al. [93] in a similar study found no such difference.

In this study, the SR response to heights was undoubtedly the strongest one measured. We suggest that this is due to the necessary involvement of conscious processing and emotional and cognitive experience engaged in assessment of the threat posed by heights. The amygdala is generally one of the crucial nodes in processing fear [101] and is also highly associated with skin conductance response [102,103]. Other involved regions are further responsible for contextual evaluation of the threat [104] and emotional regulation [105]. The emergence of fear of heights relatively late in human ontogeny [106,107] (in contrast to snakes – see, e.g., [108,109]) suggests some kind of maturation processes of CNS or an important component of learning are at play. In mice, a specific population of neurons named high-place fear neurons in the basolateral amygdala is responsible for "fear of heights" and this population does not respond to other fear-related stimuli [110]. The mouse basolateral amygdala as a whole is responsible for memory consolidation and processing of contextual information related to cue signals during fear conditioning learning [111]. Thus, it seems that at least in rodents, the proximate mechanism of "fear of heights" accounts for contextual cues and may connect innate fear responses with memories representing previous fear experiences with heights. In our experiment, heights were also perceived in a strongly contextual way, as indicated by the highest fear ratings of images with especially pronounced perspective, and corresponding high SR responses. To summarize, the threat of heights is an ancestral threat stimulus, but with an important component of conscious cognitive processing and learning across the lifespan.

Of course, responses to the threat of snakes can also be modified by context and personal experience. However, such modifications may represent only an extension of an otherwise straightforward avoidance response, rather than a core component of the threat's cognitive appraisal. In our recent study [17], snake stimuli were associated with comparatively low activation of the prefrontal cortex and other regions involved in context representation and memory retrieval, such as the parahippocampal gyrus. Unfortunately, that study did not include height-related stimuli, and fMRI research on this topic remains sparse. Nevertheless, an interesting insight is provided by a study of acrophobic patients and healthy controls [112]. The authors reported a substantial decrease in functional connectivity in the fusiform gyrus, parahippocampal gyrus, superior frontal gyrus, caudate nucleus and cuneus—regions involved in comparing current and past experiences, and in generating and evaluating alternative action plans. It should be emphasised that this pattern does not apply to all specific

phobias. For example, sensorimotor, frontoparietal and visual regions showed the largest differences relative to controls, suggesting impaired coupling between perceptual and action-related regions and higher-order control regions in two recent studies on this topic [113,114]. This leads us to conclude that even all ancestral threats need not to be processed the same in terms of fear circuit. Moreover, the assumption that stimuli from certain categories, such as snakes, should consistently evoke the strongest responses across all tasks and physiological measures, followed by a predictable hierarchy of other stimuli seems to be an oversimplification.

Substantial involvement of brain regions associated with context and prior experience was observed in the above-mentioned fMRI study [17] for both firearms and airborne disease stimuli. While we observed a significant relationship between subjective fear intensity and SR response probability for these stimulus categories (Figs 5 and 6), this relationship was not as strong as for heights. One might therefore ask why context dependency appears so pronounced for the threat of heights, yet only modest for firearms and airborne diseases. Although we can only speculate, we suggest that personal experience may represent a second key factor. It is reasonable to assume that all participants had personal experience with a wide range of height-related scenarios, enabling them to process each stimulus independently and accurately assess its specific cues. By contrast, firearms are relatively uncommon in the Czech Republic, and participants may have had little personal experience with this type of threat, leading them to rely more on broad cues and general categorization. Finally, the threat of an airborne pandemic was universally experienced by participants due to the recent COVID-19 pandemic. However, the measurements were conducted towards the end of the pandemic, when most of the population had become habituated to many of the cues used in the study. This may again indicate that, despite drawing on personal experience, participants relied largely on broad categorization of available cues.

## Limitations and future directions

The study's methodology has three main limitations. First, presenting several categories of threat stimuli in a short period may create artificial overlays, as responses to different stimuli can interact. This makes unambiguous interpretation of psychophysiological indicators of emotional states difficult. We addressed this issue by adopting a between-subjects design in which each participant was exposed to only one category of threat stimuli, along with the same control condition. This enabled us to test a sufficient number of stimuli within each category and confirm that responses were not driven by the choice of a particular stimulus. However, this advantage comes at a cost: differences in emotional responses between stimulus categories may be affected by participants' individual emotional sensitivity to specific types of stimuli. Although we controlled for emotional sensitivity using psychometric instruments, this limitation could be fully resolved only by conducting a within-subjects study with more limited number of stimuli per category. Such approach could be pursued in future research, using stimuli whose reliability has been validated in the present study.

Second, changes in skin resistance—the basis of the psychophysiological measures used in this study—do not occur within tens or hundreds of milliseconds, as is typical for brain responses measured through evoked potentials. Instead, electrodermal responses appear with a certain and variable delay. Most studies agree that after a stimulus is presented, the corresponding phasic change in skin conductivity occurs roughly one to three seconds after stimulus onset [27]. Because our experiment focused on the phasic component of skin resistance (that is, detecting peaks that indicate stimulus-evoked changes), the delay of electrodermal response can only be captured as differences in response latencies across stimulus categories within the 1–5 s analysed window (see Fig 4). Although this study was not sufficiently powered to compare this temporal component across categories, we hope that our findings may inspire future research to examine these dynamics in more detail.

Third, because the stimuli were not presented subliminally, it is difficult to disentangle automatic from conscious components of the emotional response. In this study, the influence of unconscious and conscious processing on responses to ancestral versus modern threats can be inferred only by comparing the psychophysiological reaction to each stimulus with participants' subjective (conscious) evaluations of that same stimulus. Where subjective ratings do not account for the psychophysiological response, a stronger contribution of unconscious processing may be inferred. Although unconscious responses can in principle be measured using backward masking, this technique requires stimuli to be presented for very

brief intervals. Such short presentations are not directly comparable with the longer stimulus durations used in our paradigm and would introduce a different set of limitations. Nevertheless, other research is necessary to help clarify whether the autonomic nervous system distinguishes between these categories or whether both types can activate comparable defensive reactions depending on their perceived relevance and potential for harm.

## Conclusions

In this study, we aimed to compare skin resistance (SR) response, subjectivity evoked fear, and their mutual relationship in several types of ancestral and modern threat stimuli. We found that SR response was generally the strongest towards stimuli simulating the threat of heights, albeit it was also highly context-dependent with the SR response more than twice as likely in stimuli eliciting very high fear than stimuli eliciting no fear. No difference between SR response towards venomous snakes and firearms clearly demonstrated that SR response does not binarily depend on evolutionary relevance of the fear stimulus. Subjectively, venomous snakes were rated as the most fear-eliciting category of stimuli, yet the conscious rating of individual stimuli was not related to SR response to them. Instead, specific sensitivity to fear of snakes – a stable personality trait – correlated with mean SR response towards snake stimuli. Overall, this study convincingly shows that responses may differ tremendously depending on context even for threats which are generally considered ancestral.

## Supporting information

**S1 Fig. Histograms of experimental stimuli rating, separately for each threat category.**
(XLSX)

**S1 File. Theoretical background on selection of stimuli categories.**
(XLSX)

**S2 File. Modified version of ATGV questionnaire.**
(XLSX)

**S3 File. K-cluster analysis of individual stimuli.**
(XLSX)

**S1 Table. Sources of experimental stimuli photos.**
(XLSX)

**S2 Table. Data, part 1.**
(XLSX)

**S3 Table. Data, part 2.**
(XLSX)

**S4 Table. Effect of stimulus category on skin resistance response.**
(XLSX)

**S5 Table. Effect of stimulus category, elicited fear, and their interaction on probability of skin resistance response.**
(XLSX)

**S6 Table. Effect of stimulus category, presence of skin resistance (SR) response, and their interaction on fear rating.**

 

(XLSX)

**S7 Table. Results of Spearman rank order correlation tests between the questionnaires scores and average amplitude of skin resistance (SR) change.**
(XLSX)

## Author contributions

**Conceptualization:** Daniel Frynta, Eva Landová.

**Data curation:** Iveta Štolhoferová, Tereza Hladíková, Šárka Peterková.

**Formal analysis:** Iveta Štolhoferová, Tereza Hladíková.

**Funding acquisition:** Eva Landová.

**Investigation:** Tereza Hladíková, Markéta Janovcová.

**Methodology:** Markéta Janovcová, Šárka Peterková.

**Project administration:** Markéta Janovcová.

**Software:** Šárka Peterková.

**Supervision:** Daniel Frynta, Eva Landová.

**Visualization:** Iveta Štolhoferová.

**Writing – original draft:** Iveta Štolhoferová, Tereza Hladíková, Eva Landová.

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
