## [Decision Letter · Decision Letter 0]

13 Oct 2025

Dear Dr. Landová,

Thank you for submitting your manuscript to PLOS ONE. After careful consideration, we feel that it has merit but does not fully meet PLOS ONE’s publication criteria as it currently stands. Therefore, we invite you to submit a revised version of the manuscript that addresses the points raised during the review process.

The reviewers provide complementary feedback. Reviewer 1 recommends minor revisions focused on structure and clarity, whereas Reviewer 2 raises methodological and analytical severe issues requiring a relevant revision. The authors should carefully address Reviewer 2’s concerns on design, analysis, and interpretation while incorporating Reviewer 1’s suggestions to improve clarity and completeness.

We look forward to receiving your revised manuscript.

Kind regards,

Alberto Greco

Academic Editor

PLOS ONE

Journal Requirements:

3. Please include captions for your Supporting Information files at the end of your manuscript, and update any in-text citations to match accordingly. Please see our Supporting Information guidelines for more information: http://journals.plos.org/plosone/s/supporting-information .

4. Please remove all personal information, ensure that the data shared are in accordance with participant consent, and re-upload a fully anonymized data set.

Reviewers' comments:

Reviewer's Responses to Questions

**Comments to the Author**

1. Is the manuscript technically sound, and do the data support the conclusions?

Reviewer #1: Yes

Reviewer #2: Partly

2. Has the statistical analysis been performed appropriately and rigorously?

Reviewer #1: Yes

Reviewer #2: N/A

3. Have the authors made all data underlying the findings in their manuscript fully available?

Reviewer #1: Yes

Reviewer #2: Yes

4. Is the manuscript presented in an intelligible fashion and written in standard English?

Reviewer #1: Yes

Reviewer #2: Yes

Reviewer #1: The paper describes a study on the physiological reactions to ancestrally- vs culturally-feared stimuli. The sample is larger than the average of this research line; all procedures and data are exhaustively presented and shared; the introduction and discussion are in general coherent with the results. Broadly speaking, the article is of good quality, and aims at disambiguating a research question which led to controversial findings.

Despite the generally-good quality of the paper, I have some concerns about the Introduction and the Discussion sections:

• the introduction is quite verbose due to an exhaustive specification of the nature/nurture evidences concerning each of the stimulus categories (i.e., snakes, guns...). While I agree with the necessity to elucidate such information, I find it too long and specific to be written in an introduction: the decisional process which led to choose these stimuli is rightfully pertinent to the introduction, but such level of detail is not. Thus, the solution I propose is to transfer in a supplementary table or chapter all the detailed information about all the studies supporting the nature/nurture debate on each stimulus category, leaving in the introduction few more general sentences summarizing the choice of some ancestrally-feared and of some culturally-feared stimuli. On the other hand, the space saved in the introduction should be partially filled with some little discussion of the scientific debate on subliminally-induced skin conductance reactions in response to the same kinds of stimuli (ancestral or cultural): this is only slightly mentioned so far, while I would find it very relevant to introduce (and, later, discuss) your results. Indeed, the authors did a great job recruiting 119 participants and showing them this variety of pictures, but these results alone cannot give a totally exhaustive answer about the nature/nurture picture processing debate. This leads me to the concerns about the Discussion;

• coherently with the results, it is discussed that there is not much difference between ancestral and cultural picture processing, with the only exception of height pictures. However, why heights and not – e.g. – snakes? They are both ancestrally-feared stimuli that should be expected to elicit a comparable reactivity: please detail this better. More importantly, these results concern the skin reactions to pictures lasting many seconds: during this time, lot of things could happen. For example, an initial reaction could be stronger in the first (milli)seconds for ancestral stimuli, while a latter thinking on culturally-feared stimuli could elicit a delayed reaction: this would make a difference at the beginning still yielding no significant differences on a larger window. It is not clear to me whether this possibility can be unambiguously excluded based on the statistical analyses, which seems to not deeply analyze temporal components. If you did not address the temporal facets of skin reaction, this is still ok but should be mentioned as a potential limitation while discussing the results. Indeed, to really exclude the existence of a difference between the processing of ancestrally-feared and culturally-feared stimuli, you should also have administered the same pictures subliminally. This was not probably feasible in the timing of the study, and it is ok to present only results about clearly-visible stimuli: but should nevertheless described as a potential limitation in the answer to the research questions of interest, and suggested as a future development of the study. This issue also concerns the relatively-simple analysis performed on skin conductance data, which is ok but could be deepened with more exhaustive approaches: this could be also mentioned as a potential limitation and as an invitation for other researchers to perform other analyses on the shared data.

Other minor issues concern:

• the full availability of data. I really appreciated the data shared by the authors in form of supplementary tables, however I would like them to upload everything (including both the raw and pre-processed electrodermal data, and the R/Matlab scripts used to analyze them) on some open repository (e.g., Open Science Framework) for a better replication of the study and re-analysis of its results;

• some typos of minor importance (e.g., uppercase after semicolon in line 295) which nevertheless should be carefully checked and corrected. Also, I see the figures with quite a low quality, which is still intelligible but I hope that the final version will be of better quality.

After addressing all these points (including the full sharing of data), I will see no reasons to not endorse this interesting paper.

Reviewer #2: In this work, Štolhoferová, et al. aim to elucidate physiological responses through skin conductance on stimuli representing ontogenetic and phylogenetic threats. Additionally, they investigate the bilateral relationship between physiological responses and subjective measurements. I appreciate this interesting avenue of research and the efforts of the team involved to acquire over a hundred subjects in a few months. However, I have serious doubts regarding the experimental setup, the rigour, and suitability of statistical analysis. I think the manuscript needs to be thoroughly revised to highlight the contextual importance of stimuli, rather than finding a dichotomy between ancestral and modern threats. Specifically, the fact the even snakes – a canonical “fearful” stimulus – does not elicit a strong response, suggests contextual processes are ongoing. I think the manuscript would have to argue along the lines of such selective embodiment.

MAJOR POINTS

1. The major aim of the study seems to be the differentiation of particular categories of threats (ancient vs modern) based on skin conductance responses. Most stimuli included are questionably attributable to either class. For instance, even though “guns” are a modern concept (and thus a learned association with danger), the underlying circuits that drive the signal-of-interest may still be rooted in evolutionary mechanisms. As such, babies seem to be quite curious around snakes, suggesting there is a similar learned association. Similar arguments can be made for the other stimuli, rendering a hard dichtomy between ancient and modern threats rather difficult. All stimuli, therefore, probably activate the same domain-general salience network in the brain. It will care about how close the threat is, how probable damage to the body is, and how controllable the threat is.

2. Given that personal, cultural, and social factors play such a large role in shaping the fear response, I do not think a between-subjects design is the most appropriate. I do understand that between-subjects design results in less carry-over, but I think controlling for individual differences outweighs the possibility that responses to one stimulus might affect the next. This might also underlie the result that only heights seem to really induce a physiological response.

3. The statistical analyses mainly depend on GLMMs and AICs. In light of the obscure nature of the stimuli, I would have like to see clustering approaches to identify how stimuli relate to one another. Now the reader is presented with what seems to be an “odd” result (only height seems to drive a response). Specifically, the fact that snakes elicited such a weak response should have prompted more in-depth analyses. Rather than digging in further into how this may have arisen, the reader is presented with a broad “what even is fear” discussion.

4. The introduction/discussion in general are fragmented: each stimulus type received a paragraph as to why the obtained results could have made sense. I am missing a discussion on the overlap of stimuli, shared circuitry, and how personal/contextual cues might shape the responses.

MINOR POINTS

1. I would have liked a more in-depth discussion on how the skin resistance signal arises. This is now briefly touched upon in line 47-49, but could be expanded to aid the novices to skin condutance. Moreover, the authors may want to strengthen the justification for EDA over other modalities that are linked to arousal (e.g., pupil dynamics, heart period responses, etc), or even combine them (in future studies).

2. The visualizations of this study are rather sparse. I am only getting to see summary/descriptive statistics, but no actual skin conductance traces. This removes me far from the actual data.

3. In Figure 2b, the different letters “indexes unsignificant differences between categories”. The table in Table S5b shows that the leaves and airborne disease categories do not significantly differ. Why, then, do they have different indices? Am I looking in the wrong table? Or is “leaves” always annotated with “a” as it’s the control condition?

4. Figure 3 requires uncertainty measures around the fit. It is unlikely that the distribution of rating values is homogeneous across ratings (due to central tendency or floor effects) (See Figure S1). The extreme linearity is hard to interpret without a sense of variability.

5. Line 143 reads “Nonetheless, it is not straightforward as it might seem to estimate how long airborne diseases have …”. I fail to see why this would be straightforward.

6. Line 231: it is easier to just refer to “participants”

7. Line 241: the stimuli alternated? Does this mean there was an absence of randomization? This would induce quite some predictive processes going on, as the stimulus sequence is predictable.

8. Line 260: not a critique, but 18-74 is a broad age range!

9. Line 261: did the power analysis also include the between-subjects structure of the experiment?

10. Line 291: given the potential bottom-up versus top-down influences to different stimuli, it would actually be interesting to say something about the response latencies. Also with respect to the non-random nature of the experiment, one may expect responses before the stimulus onset.

11. Given the subtle effects that seem to be at play, I would like to see more uncertainty measures (if AIC) produces something like that (e.g., confidence intervals, Cohen’s D).

12. I think figure 4 requires more discussion as to what it implicates. Which ratings drive the effect? A strong reduction of rating = 1, but where do these other ratings go? Uniformly to the other score or generally towards higher ratings?

13. In section 3.5, did you control for the number of tests (i.e., number of questionnaires) or were they all considered to be independent samples?

14. Line 537: can you show this interstimulus separation?

15. Grammatical/sentence structure is sometimes a bit complex:

a. Line 99: Restructure to “… these stimuli [40,41]. For example, …”

b. Lines 104-107 are hard to parse

c. Lines 127-130 are also hard to parse

d. Line 473: “bring” should be “brought”

e. Line 512: “though” should be “through”

f. I may have missed other typos

16. I applaud the authors for sharing all the data. However, it would be great if the accompanying code could be shared to. This should include documentation on how to use the code, so other researchers (including me) can learn more from this data.

17. After manuscript revision, the authors may reconsider the title of the manuscript. It now focuses on the fact that among evolutionary responses, only heights evokes a strong response compared to snakes. I think the title should go more in the direction of unified/selective embodied representations of this fear-spectrum.

TECHNICAL SUGGESTIONS

1. For clustering approaches, one could think of k-Means clustering or support vector machines to try and elucidate particular clustering of stimulus types. This may either show an expected evolutionary progression, or it may be inseparable, which would be evidence for a domain-general mechanism.

2. As salience is such an important factor in studies like this, it may be valuable investigating the stimulus energy of the presented images across stimulus types. This may say something about how much each stimulus captures attention.

CONCLUDING REMARKS

Again, I would like to commend the authors for their hard work and I apologize if my tone was harsh. With a major revision of the framework, updated visualizations, and additional statistical tests, I feel this work can provide another piece of the puzzle. Specifically, in the sense that even the threat of “ancestral” stimuli are learned, and their responses may differ tremendously depending on context.

**Do you want your identity to be public for this peer review?** For information about this choice, including consent withdrawal, please see our Privacy Policy

Reviewer #1: **Yes:** Sergio Frumento

Reviewer #2: No

---

## [Author Response · Author response to Decision Letter 1]

28 Nov 2025

The responses can be found in the text document named Review_PLoS_Physiology.docx.

---

## [Decision Letter · Decision Letter 1]

2 Feb 2026

Dear Dr. Landová,

Thank you for submitting your manuscript to PLOS ONE. After careful consideration, we feel that it has merit but does not fully meet PLOS ONE’s publication criteria as it currently stands. Therefore, we invite you to submit a revised version of the manuscript that addresses the points raised during the review process.

The reviewer raises only minor points. Please revise the concluding sentence to better reflect the study’s implications, standardize figure aesthetics and resolution, clarify the dashed lines in Fig. 5, and comment on whether advanced EDA processing approaches were considered.

Please submit your revised manuscript by the Mar 19 2026 11:59PM If you will need more time than this to complete your revisions, please reply to this message or contact the journal office at plosone@plos.org . A letter that responds to each point raised by the academic editor and reviewer(s). You should upload this letter as a separate file labeled 'Response to Reviewers'.A marked-up copy of your manuscript that highlights changes made to the original version. You should upload this as a separate file labeled 'Revised Manuscript with Track Changes'.An unmarked version of your revised paper without tracked changes. You should upload this as a separate file labeled 'Manuscript'.

We look forward to receiving your revised manuscript.

Kind regards,

Alberto Greco

Academic Editor

PLOS One

Journal Requirements:

Reviewers' comments:

Reviewer's Responses to Questions

**Comments to the Author**

Reviewer #1: All comments have been addressed

Reviewer #2: (No Response)

2. Is the manuscript technically sound, and do the data support the conclusions?

Reviewer #1: Yes

Reviewer #2: Yes

3. Has the statistical analysis been performed appropriately and rigorously?

Reviewer #1: Yes

Reviewer #2: Yes

4. Have the authors made all data underlying the findings in their manuscript fully available?

Reviewer #1: Yes

Reviewer #2: Yes

5. Is the manuscript presented in an intelligible fashion and written in standard English?

Reviewer #1: Yes

Reviewer #2: Yes

Reviewer #1: I'm satisfied with the changes made by the authors to meet my suggestions and with the motivations presented to justify the few occasions in which my suggestions have not been fully accepted.

I'm only concerned about a last point that I encourage to address (even if I do not consider it critical enough to prevent endorsement). In the last sentence of Conclusions (lines 732-734) is indeed written that "this is still one of the first (if not the very first) study comparing electrodermal responses across different categories of evolutionary versus modern threat stimuli": while it is one of the few studies accounting for this comparison, this is clearly not the "very first", as the authors themselves cited some articles previously comparing ancestral VS modern threats (Hugdahl & Johnsen, 1989; Flykt, Esteves & Öhman, 2007; Carlson et al., 2009; Luck et al., 2020). While the present paper integrates and enriches this research line, its "very first" nature sounds like an unnecessary overstatement: my suggestion is to entirely remove this last sentence, or at least the "(if not the very first)" specification.

Reviewer #2: I would like to thank and commend the authors for their rigorous work I believe many comments have been appropriately dealt with and the manuscript reads well now. I only have a few minor points that the authors could consider, but they do not necessarily prevent the manuscript from being suitable for publication.

1. I find the last sentence of the conclusion (starting line 618) somewhat negative. Every study has limitations, so I would prefer a concluding sentence with more body. What are the implications of this study, how does it progress science, how does it tie in with existing literature. Maybe something similar to my suggestion in the "concluding remarks" section of my previous review if that summarizes the study succintly.

2. The figures seem a bit disjointed; there are differences in axis width, font size, and other aesthethic elements across the figures. It would be great if the figures could be homogenized through e.g. python (matplotlib) or R to ensure the visual elements and scaling of font sizes are the same across figures. It also seems that the resolution is quite low, but that may be because of the initial submission. If not, I would recommend generating high-resolution figures.

2a. In figure 5, is there a reason that only the snakes are denoted with dashed lines rather than solid lines?

3. The current implementation of the EDA processing seems to rely on peak-scoring methods. Have any other, more advanced processing pipelines such as psychophysiological modelling been considered to extract response estimates for the different stimuli?

PS: Apologies for the delay.

**Do you want your identity to be public for this peer review?** For information about this choice, including consent withdrawal, please see our Privacy Policy

Reviewer #1: **Yes:** Sergio Frumento

Reviewer #2: No

---

## [Author Response · Author response to Decision Letter 2]

9 Feb 2026

Thank you for a possitive feedback. Please see the attached file for detailed responses.

---

## [Editor Report · Decision Letter 2]

10 Feb 2026

Subjective and psychophysiological response to pictures of ancestral and modern threats: not all evolutionary threats are alike

PONE-D-25-24719R2

Dear Dr. Landová,

We’re pleased to inform you that your manuscript has been judged scientifically suitable for publication and will be formally accepted for publication once it meets all outstanding technical requirements.

Kind regards,

Alberto Greco

Academic Editor

PLOS One
---

## [Editor Report · Acceptance letter]

PONE-D-25-24719R2

PLOS One

Dear Dr. Landová,

I'm pleased to inform you that your manuscript has been deemed suitable for publication in PLOS One. Congratulations! Your manuscript is now being handed over to our production team.

Kind regards,

on behalf of

Dr. Alberto Greco

Academic Editor

PLOS One